# Visually-Situated Natural Language Understanding with Contrastive Reading Model and Frozen Large Language Models

**Geewook Kim**[1,2*]    **Hodong Lee**[1,3*]    **Daehee Kim**[1*]    **Haeji Jung**[3*†]    **Sanghee Park**[1*]
**Yoonsik Kim**[1*]    **Sangdoo Yun**[4‡]    **Taeho Kil**[1‡]    **Bado Lee**[1‡]    **Seunghyun Park**[1‡]

[1]NAVER Cloud AI    [2]KAIST AI    [3]Korea University    [4]NAVER AI Lab

## Abstract

Recent advances in Large Language Models (LLMs) have stimulated a surge of research aimed at extending their applications to the visual domain. While these models exhibit promise in generating abstract image captions and facilitating natural conversations, their performance on text-rich images still requires improvement. In this paper, we introduce Contrastive Reading Model (Cream), a novel neural architecture designed to enhance the language-image understanding capability of LLMs by capturing intricate details that are often overlooked in existing methods. Cream combines vision and auxiliary encoders, fortified by a contrastive feature alignment technique, to achieve a more effective comprehension of language information in visually situated contexts within the images. Our approach bridges the gap between vision and language understanding, paving the way for the development of more sophisticated Document Intelligence Assistants. Through rigorous evaluations across diverse visually-situated language understanding tasks that demand reasoning capabilities, we demonstrate the compelling performance of Cream, positioning it as a prominent model in the field of visual document understanding. We provide our codebase and newly-generated datasets at
https://github.com/naver-ai/cream.

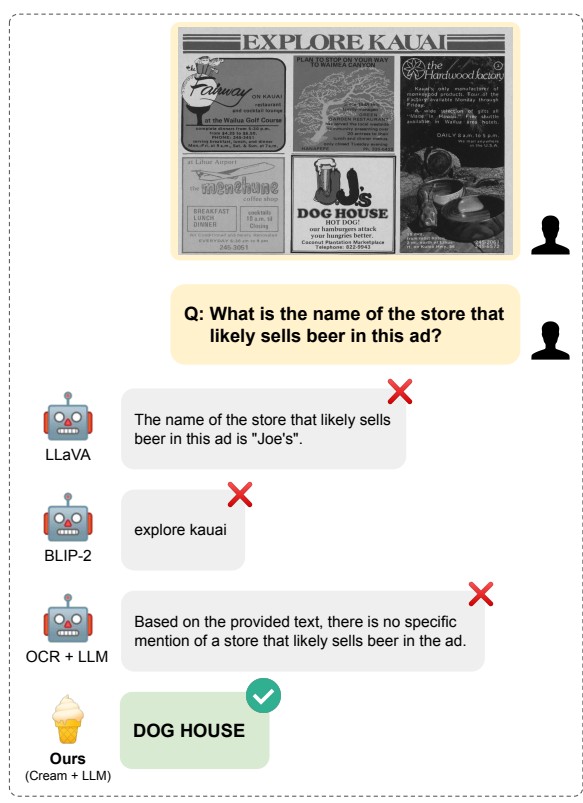

**Figure 1: Comparison on a text-rich image.** The proposed method, Cream, precisely interprets and reads the relevant store's name from a poster containing multiple text instances and visuals, overcoming limitations of existing approaches (e.g., OCR+ChatGPT). Our Cream efficiently extracts visual features from the image, thus enabling LLMs to provide an appropriate response.

## 1 Introduction

Recent advances in Large Language Models (LLMs) (Brown et al., 2020; OpenAI, 2023; Zhang et al., 2022; Touvron et al., 2023) have facilitated the development of numerous real-world applications. Researchers are increasingly focusing on extending these unimodal LLMs to multimodal LLMs, particularly Large Visual Language Models (LVLMs), leveraging vision encoders designed to tackle information-rich visual tasks (Radford et al., 2021a; Tsimpoukelli et al., 2021; Wang et al., 2022a; Alayrac et al., 2022; Wang et al., 2022b; Driess et al., 2023; Zhu et al., 2023).

To evaluate LVLMs, various downstream tasks have been employed, such as image captioning, visual dialogue, grounding, reasoning, and question generation. Although LVLMs demonstrate impressive results in these tasks, a recent study (Liu

---

* Core contributions. Correspondence to Geewook Kim: gwkim.rsrch@gmail.com.

† Work done during internship at NAVER Cloud AI.

‡ Advisory contributions. A description of each author's contribution is available at the end of this paper.

et al., 2023b) argued that LVLMs exhibit limitations when dealing with visual tasks on text-rich images, leading to reduced applicability in real-world applications, such as Document Visual Question Answering (Document VQA). Document VQA tasks require the comprehensive analysis of multiple types of information, including text, objects (e.g., figures or charts), and layout. However, existing LVLMs struggle to deliver satisfactory solutions due to their limited ability to extract fine-grained features from images, as shown in Figure 1.

In this paper, we introduce **Cream**, *Contrastive reading model*, specifically designed to effectively overcome these limitations. Cream features a streamlined and practical architecture that seamlessly integrates a general-vision encoder with auxiliary encoders and novel training techniques. In addition to a primary vision encoder for overall visual feature extraction from document images, Cream employs auxiliary encoders—such as OCR and object detectors—for text and object-specific feature extraction. Cream handles fine-grained features without missing image details while understanding the visual context. When combined with LLMs, Cream overcomes the limitations of LVLMs and achieves robust performance on text-rich images. To further enhance the model, we propose a contrastive feature alignment method to balance disparities between the vision and auxiliary features extracted from each encoder during training.

We conduct extensive experiments on challenging text-rich Document VQA benchmarks. We perform experiments on two models: our standalone Cream model and a model that combines our Cream model with frozen LLMs. The experimental results demonstrate that standalone Cream achieves results comparable to the state-of-the-art in tasks that necessitate the extraction of specific text information from document images. Furthermore, we observe that when combined with LLMs, Cream demonstrates robust performance in Visual Document Understanding (VDU) tasks, which are challenging for existing LVLMs. Lastly, we will open-source Cream's codebase and the newly built VQA datasets to foster further research and innovation in the field of visual document understanding.

Our contributions are summarized as follows:

- We present a novel neural architecture and associated model training techniques tailored for text-rich image understanding tasks, highlighting improved performance across challenging VDU tasks.

- We suggest a contrastive learning-based training technique to improve performance and enhance robustness.

- We offer an accessible approach that allows for integrating the proposed model into LLMs, thus expanding the versatility of LLMs in multimodal domains by providing rich visual information.

- We demonstrate the proposed method's superior performance on challenging text-rich VDU benchmarks via rigorous experiments.

- We contribute valuable resources to facilitate ongoing research and development in VDU tasks by sharing our codebase and newly-generated datasets.

## 2 Related Work

### 2.1 Applying LLMs to Visually-Situated NLU

Large language models (LLMs) have demonstrated outstanding performance across various applications (Rae et al., 2021; Brown et al., 2020; Chowdhery et al., 2022; Hoffmann et al., 2022; Touvron et al., 2023), demonstrating their ability to generate responses in accordance with user intent (Ouyang et al., 2022; Shahriar and Hayawi, 2023). Researchers have explored the use of LLMs in tasks related to visual understanding (Tsimpoukelli et al., 2021; Alayrac et al., 2022; Zhu et al., 2023; Liu et al., 2023a; Ye et al., 2023); however, their application in visually-situated natural language understanding (NLU) tasks, such as question answering on text-rich document images, remains limited. Previous approaches have attempted to integrate visual embeddings or OCR tokens into LLMs to address this gap (Li et al., 2023; Dai et al., 2023; Wang et al., 2023). However, these methods suffer from inefficiency, as they require numerous LLM tokens and entail considerable computational overhead. For instance, documents in DocVQA (Tito et al., 2021) consume an average of 400 and a maximum of nearly 3K OCR tokens. To overcome these obstacles, we propose the integration of Cream, which imparts language-image understanding capabilities to LLMs using soft visual prompts with a fixed size. This integration has the potential to enhance the efficiency and accuracy of LLMs in visually-situated NLU tasks.

## 2.2 Visual Document Understanding

Visually-situated NLU blends computer vision and NLU techniques to accurately analyze visual data through language. Early approaches emphasized OCR and leveraged extracted text for contextual analysis (Xu et al., 2020; Hong et al., 2022). Some contemporary methods directly processed document images, circumventing the need for external OCR models (Kim et al., 2022; Davis et al., 2023; Lee et al., 2022; Liu et al., 2022), while others leveraged both image and OCR-extracted text for improved performances (Kil et al., 2022; Tang et al., 2022; Appalaraju et al., 2021; Xu et al., 2021; Huang et al., 2022). LayoutLMv3 (Huang et al., 2022) introduces the word-patch alignment technique, which classifies the relationship between text and corresponding image patches. UDOP (Tang et al., 2022) employs a unified encoder that handles both image and text features, transforming the information from both modalities into vision-text embeddings by summing the image patch and text features. This strategy ensures alignment between modalities, such as image patches, tokens, and layout information, by fusing multimodal features within a single encoder. In contrast, Cream extracts fine-grained, aligned multimodal features through contrastive learning (CL), eliminating the need for the fusion encoder. The CL approach prevents over-fusion of information from each modality, allowing the decoder to effectively utilize the multimodal semantics inherent in visually-rich documents. Moreover, due to this architectural design, Cream exhibits enhanced robustness to OCR errors.

## 3 Method

Our goal is to develop a system that accurately answers natural language questions based on specific evidence in an input image. To achieve this, we propose a model that explicitly identifies feature evidence in the image, such as texts and objects.

## 3.1 Contrastive Reading Model

We introduce *Contrastive reading model* (**Cream**) with two potential application scenarios: (i) functioning independently, where a decoder module of Cream directly generates a desired text information, and (ii) operating in conjunction with an LLM, serving to provide soft visual prompts for the LLM. A comprehensive overview of the entire pipeline is depicted in Figure 2.

### 3.1.1 Architecture

Cream is composed of two encoders and one decoder. The vision encoder computes vector representations of the input image patches. Concurrently, feature evidence, such as text or general object information in target images, is extracted in text format and encoded by an auxiliary text encoder. The embeddings from both encoders are then aligned using our proposed CL scheme (see Figure 2). The aligned features undergo further processing by the decoder to extract necessary information. Additional details will be provided in the sections that follow.

**Vision Encoder**  The vision encoder converts the input image $\mathbf{x} \in \mathbb{R}^{H \times W \times C}$ into embeddings $\{\mathbf{z}_i | \mathbf{z}_i \in \mathbb{R}^d, 1 \leqslant i \leqslant n\}$, where $n$ is the feature map size or the number of image patches, and $d$ is the dimension of the output vectors. We can use CNN-based models (He et al., 2016) or Transformer-based models (Dosovitskiy et al., 2021; Liu et al., 2021) as the encoder network. In this study, we employ Vision Transformer (Dosovitskiy et al., 2021) with a 2D absolute position encoding (Xu et al., 2020) and a variable-resolution mechanism (Lee et al., 2022) to simplify the process. The variable-resolution mechanism ensures a constant number of patches without distorting the original image aspect ratio.

**Auxiliary Encoder**  The auxiliary encoder encodes the extracted feature evidence, such as OCR boxes and general object boxes, into embeddings $\{\hat{\mathbf{z}}_i | \hat{\mathbf{z}}_i \in \mathbb{R}^d, 1 \leqslant i \leqslant \hat{n}\}$, where $\hat{n}$ is the maximum sequence length of the encoder. In Figure 3, the feature evidence is converted into token embeddings using the recognized text for the OCR boxes and the recognized semantic object label for the general object boxes. A type embedding is added to differentiate between OCR and general object boxes, and a 2D absolute position encoding is applied for encoding the location information. We employ the encoder of BART (Lewis et al., 2020) as the Auxiliary Encoder.

**Decoder**  We employ BART decoder, which takes the output embeddings of the encoders $\{\mathbf{z}_1, ..., \mathbf{z}_n, \hat{\mathbf{z}}_1, ..., \hat{\mathbf{z}}_{\hat{n}}\}$ and generates a sequence $\mathbf{h} \in \mathbb{R}^{m \times d}$, where $m$ is the length to generate. The generated hidden states are utilized in two scenarios. (i) In the standalone scenario, we apply a linear head $\mathbf{W} \in \mathbb{R}^{d \times v}$ to obtain a token sequence $\hat{\mathbf{Y}} = \mathbf{h}\mathbf{W}$, where $\hat{\mathbf{Y}} \in \mathbb{R}^{m \times v}$ is the sequence of

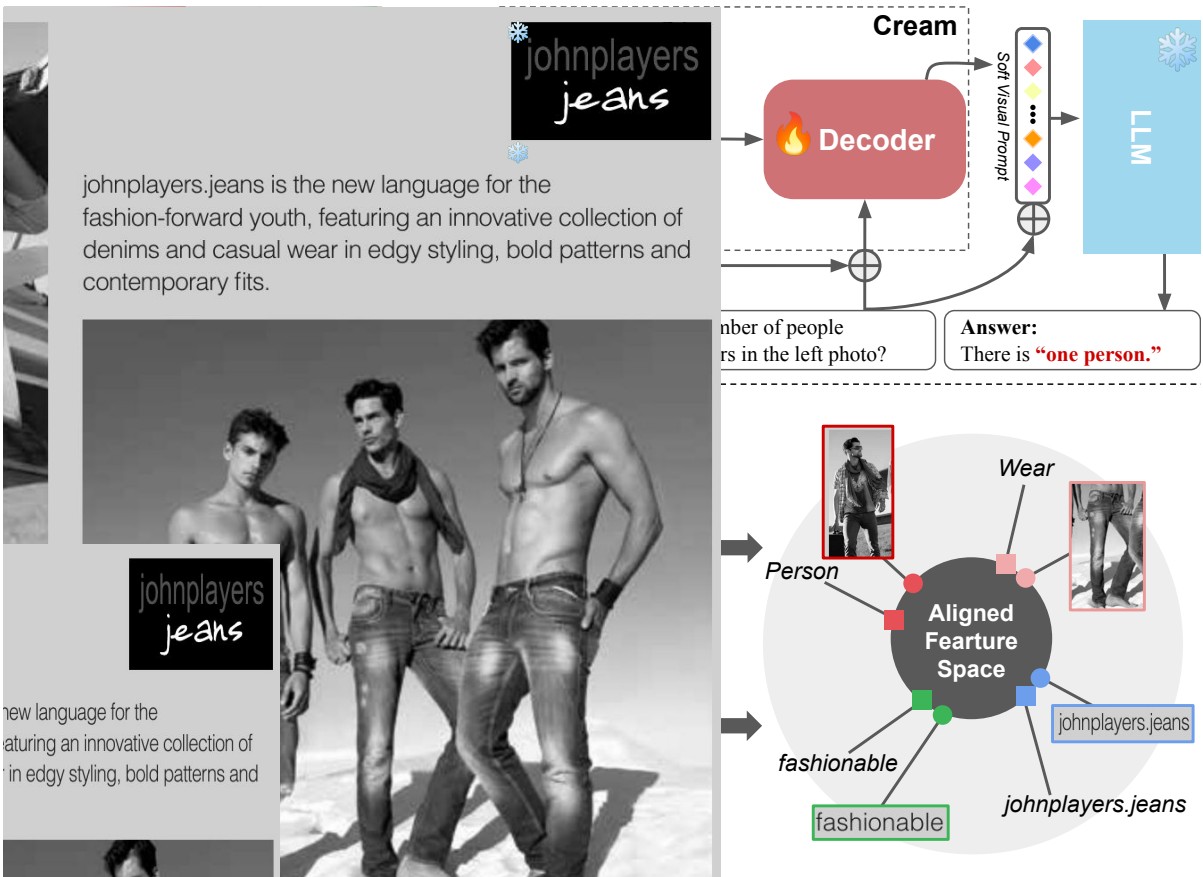

Figure 2: **Overview of Cream's Framework.** (a) Image patches are fed into the vision encoder, while information extracted from off-the-shelf detectors is processed through the auxiliary encoders if available. The encoded vectors are concatenated and then cross-attended in the decoder. The decoder, receiving both a learned query vector and a user query as inputs, serves as a soft visual prompter for the LLM. Note that the encoders are frozen during the training with LLMs. (b) Encoded vector representations are effectively aligned using a contrastive learning scheme.

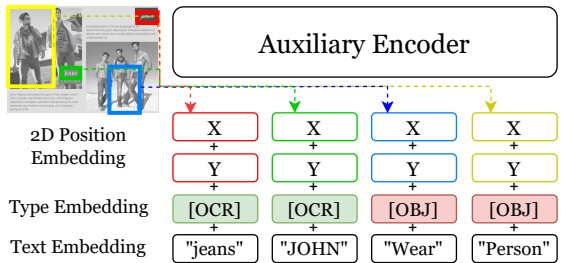

Figure 3: **Token embeddings in the auxiliary encoder.** The 2D positional embeddings are computed using the center point of each bounding box. Text embeddings are obtained through a lookup operation on a subword embedding matrix. For simplicity, words are plotted instead of subwords.

predicted tokens and $v$ is the vocabulary size. (ii) When integrated with an LLM, the hidden states are linearly mapped to be used as a soft visual prompt that combines Cream's visual understanding capabilities with the LLM's language processing abilities: $\mathbf{h}' = \mathbf{h}\mathbf{U}$, where $\mathbf{U} \in \mathbb{R}^{d \times d'}$ and $d'$ is the

LLM's input dimension. In both scenarios, we use a language modeling (LM) loss, where the model generates a target sequence of token embeddings conditioned on the image. The details of the training objective will be explained in Section 3.3.3.

### 3.1.2 Contrastive Feature Alignment

To enhance the understanding of texts and objects in images, we introduce an auxiliary encoder alongside the vision encoder (Figure 2). It is critical to note that alignment of features from these distinct encoders within a shared space is not guaranteed. As observed in our analysis in Section 4, the encoded information often suffers from misalignment, leading to a model performance degradation. To address this challenge, we introduce an efficient and simple CL strategy during the model training phase. This strategy is employed to guarantee the alignment of feature embeddings, which in turn, enhances the overall performance of the model.

As shown in Figure 3, we utilize detectors to

obtain feature evidence (e.g., texts and objects) and encode their box coordinates and text labels into vector representations. We assume that an image patch can physically overlap with certain boxes, implying that the visual patch embedding should encompass semantically relevant information similar to its corresponding auxiliary information. Given this presumption, we establish the positive relations between a visual patch embedding and its corresponding embeddings from the auxiliary encoder, while considering all other relationships as negative pairs. Our CL approach differs from conventional image-level CL methods like CLIP (Radford et al., 2021b) as our method accommodates more pairwise relations within an image, leveraging multiple available feature evidence.

For CL, we use a 2-layer Multi-Layer Perceptron (MLP) $\mathbf{f}_\theta : \mathbb{R}^d \mapsto \mathbb{R}^{d^*}$, where $d^*$ is a hyperparameter for a dimension of a common space. Most settings are similar to those of Khosla et al. (2020). More specific details are provided in Appendix A.4.1. The CL objective can be expressed as follows:

$$\sum_{(i,j)\sim P_{\text{uni}}}^{l} -\log \frac{2s(\mathbf{v}_i, \mathbf{v}_j)}{\sum_{k \in A(i,j)} s(\mathbf{v}_i, \mathbf{v}_k) + s(\mathbf{v}_j, \mathbf{v}_k)}, \quad (1)$$

where the sets $\{\mathbf{v}_i | \mathbf{v}_i \in \{\mathbf{z}_1, ..., \mathbf{z}_n\}, 1 \leq i \leq l\}$ and $\{\mathbf{v}_j | \mathbf{v}_j \in \{\hat{\mathbf{z}}_1, ..., \hat{\mathbf{z}}_{\hat{n}}\}, l + 1 \leq j \leq 2l\}$ are uniformly sampled from all positive pairs in $\{\mathbf{z}_1, ..., \mathbf{z}_n, \hat{\mathbf{z}}_1, ..., \hat{\mathbf{z}}_{\hat{n}}\}$. We refer $A(i,j)$ as $\{1, ..., 2l\}\backslash\{i,j\}$. The denominator accumulates the similarity scores over all negative pairs. The function $s(\mathbf{x}, \mathbf{y})$ is defined as $s(\mathbf{x}, \mathbf{y}) = \exp(\cos(\mathbf{f}_\theta(\mathbf{x}), \mathbf{f}_\theta(\mathbf{y}))/\tau)$, where $\cos(\mathbf{f}_\theta(\mathbf{x}), \mathbf{f}_\theta(\mathbf{y}))$ denotes computing the cosine similarity between its vector inputs, with MLP parameterized by $\theta$. The $\tau$ is the temperature parameter that modulates the softmax sharpness. The proposed CL encourages the alignment of embeddings from both encoders in the feature space, resulting in performance improvement. We validate the effectiveness of CL in our analyses (§ 4).

## 3.2 Integration of Cream and LLMs

The integration method is built upon the work proposed in BLIP-2 (Li et al., 2023), where Cream's decoder generates visual prompts for the LLM to generate text responses. We adopt the learned query mechanism from BLIP-2, which uses trainable embeddings as inputs for the Cream decoder. This allows for the generation of fixed-

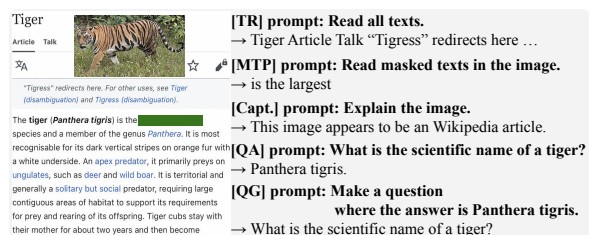

Figure 4: **Unified multitask framework.** The list of full prompts that we used is available in Appendix A.3.3.

size hidden states that the LLM can utilize. By modifying the attention mechanism to allow bi-directional flow, Cream's decoder effectively extracts visual prompts, enhancing performance in visually-situated NLU tasks.

It is worth noting that our approach differs from conventional methods that directly input visual embeddings and OCR tokens to LLMs, e.g., BLIP-2 and InstructBLIP (Li et al., 2023; Dai et al., 2023). Our method extracts relevant information from the input image and utilizes Cream's decoder to aggregate the extracted information. The generated visual prompts therefore contain OCR information, while reducing the computational overhead.

## 3.3 Model Training

### 3.3.1 Tasks

**Text Reading (TR)**  Cream reads text from top to bottom within images (Kim et al., 2022). In this task, the output of auxiliary encoder is masked to support the model learning a text reading ability.

**Masked Text Prediction (MTP)**  Cream predicts obscured characters in randomly masked OCR boxes. This task can be interpreted as an expansion of masked LM (Tay et al., 2022) to the visual domain.

**Captioning**  Cream generates captions encapsulating scene and object details, enhancing image-content understanding and object recognition.

**Question Answering (QA)**  Cream answers questions using language-image comprehension of visual contexts, emphasizing relevant image regions and textual information.

**Question Generation (QG)**  QG prompts Cream to create question sentences for provided answer texts by swapping QA components.

### 3.3.2 Unified Multitask Framework

The tasks of text reading, MTP, captioning, QA, and QG are interrelated and can be addressed using

similar approaches. They involve extracting a text sequence based on task-specific queries given an input image and its features. Our unified training framework (See Figure 4) takes prompts and images as input and generates desired answer texts for all tasks. Unlike other document understanding methods that use single task-specific prompts (Kim et al., 2022; Tang et al., 2022), Cream is trained with natural language-based prompts, facilitating seamless integration into LLMs. Our prompt is distinct to other methods such as Donut (Kim et al., 2022) and UDOP (Tang et al., 2022), which employ task-specific prompts.

### 3.3.3 Objective

Inspired by modern pre-training-and-fine-tuning strategies (Kim et al., 2022) and curriculum learning strategies (Soviany et al., 2022), we gradually increase the proportion of supervised QA data during training, initially focusing on text reading and image captioning. Two main objectives are used during training: LM and CL loss. The LM objective is to minimize a cross-entropy loss between predicted and ground truth token sequences. In line with Vaswani et al. (2017), the teacher-forcing scheme (Williams and Zipser, 1989) is employed, using ground truth as input during training for accurate contextual learning. The CL objective encourages alignment of embeddings in the feature space between vision and auxiliary encoders. The CL objective is defined in Equation 1.

The losses are combined using a weighted sum, $\mathcal{L}_{LM} + \lambda\mathcal{L}_{CL}$, where $\lambda$ is the scale factor of the CL loss. This training objective ensures effective alignment of encoded information and high performance in visually-situated NLU tasks, as demonstrated in our experiments (§ 4). When training Cream integrated with LLMs, we freeze the LLM and Cream's encoders while updating the Cream decoder via gradient descent-based training. Note that, in this phase, only the $\mathcal{L}_{LM}$ from the LLM's output layer is active.

## 4 Experiments and Analyses

### 4.1 Setups

This section provides major details on experiments. More details are discussed in Appendix A.4.

**Model Configurations** The vision encoder is initialized with LAION (Schuhmann et al., 2022) 2B pre-trained OpenCLIP (Radford et al., 2021b; Ilharco et al., 2021). The auxiliary encoder and the

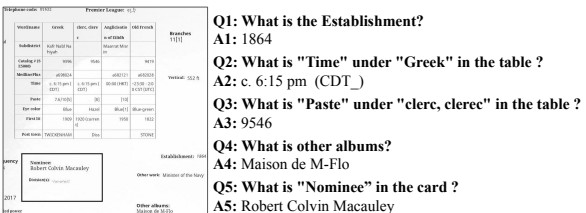

Figure 5: **Examples of synthetic VQA datasets.** Examples of other datasets are available in Appendix A.6.

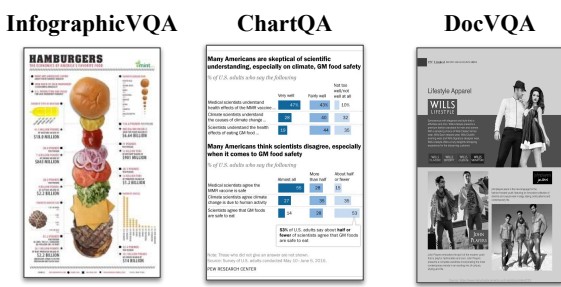

Figure 6: **Evaluation benchmarks.** We evaluate models on visual context-rich Document VQA benchmarks.

decoder modules are initialized with mBART (Liu et al., 2020). We test two model sizes: the main (18, 12, 12 layers for vision, auxiliary encoders, and decoder, respectively, with a 14×14 patch size) and a smaller ablation model (9, 6, 6 layers with a 32×32 patch size). For LLM integration tests, we use Vicuna7B (Chiang et al., 2023).

**Datasets** Table 1 provides an overview of training datasets and their statistics. For **TR and MTP**, we employ IIT-CDIP (Lewis et al., 2006) and WEBVICOB (Kim et al., 2023). WEBVICOB[1] is a visual corpus generator for a Wikipedia dump. We create a visual corpus of size 30M. For **Captioning**, we use CC3M dataset, which contains general images and accompanying text descriptions. For **QA and QG**, we introduce various supervised VQA datasets to improve Cream's visual understanding capabilities. To further boost the text-rich image understanding, using Wikipedia data source, we generate three synthetic VQA datasets: WKVVQA, SquadVQA, and TydiVQA. WKVVQA comprises synthetic document images with key-value pairs extracted from the Wikipedia. Samples of WKVVQA are shown in Figure 5. Both SquadVQA and TydiVQA expand unimodal datasets (Rajpurkar et al., 2018; Clark et al., 2020) by rendering its context page with WEBVICOB. More details on the dataset construction are available in Appendix A.5.

---

[1] https://github.com/clovaai/webvicob

| Dataset | Task | Size (#Img) |
|---|---|---|
| IIT-CDIP (Lewis et al., 2006) | TR / MTP | 11M |
| WEBVICOB (Kim et al., 2023) | TR / MTP | 30M |
| CC3M (Sharma et al., 2018) | Captioning | 3M |
| ChartQA (Masry et al., 2022) | QA / QG | 18K (train) |
| InfoVQA (Mathew et al., 2022) | QA / QG | 4K (train) |
| DocVQA (Tito et al., 2021) | QA / QG | 11K (train+val) |
| VisualMRC (Tanaka et al., 2021) | QA / QG | 9K (train+val) |
| DVQA (Kafle et al., 2018) | QA / QG | 200K (train) |
| OCRVQA (Mishra et al., 2019) | QA / QG | 146K (train) |
| STVQA (Biten et al., 2019) | QA / QG | 17K (train) |
| TextVQA (Singh et al., 2019) | QA / QG | 25K (train+val) |
| VizWizVQA (Gurari et al., 2018) | QA / QG | 15K (train) |
| VQAv2 (Goyal et al., 2017) | QA / QG | 83K (train) |
| WTQ (Pasupat and Liang, 2015) | QA / QG | 14K (train) |
| SquadVQA | QA / QG | 130K (train) |
| TydiQA | QA / QG | 4K (train) |
| WKVVQA | QA / QG | 800K |

Table 1: **Statistics of the training datasets.**

| Phase | Task Proportion |
|---|---|
| Standalone | TR, MTP, Capt., QA, QG (22, 46, 22, 5, 5) → QA (100%) |
| LLM Integration | QA (100%) |

Table 2: **Task proportions according to phases.**

**Evaluation** The models are evaluated on text-rich VQA benchmarks, including ChartQA (Masry et al., 2022), InfographicVQA (InfoVQA) (Mathew et al., 2022), and DocVQA (Tito et al., 2021) (Figure 6). While DocVQA serves as a representative text-rich benchmark, its majority of extractive QA samples demand less reasoning capability compared to the other two datasets. Both InfoVQA and ChartQA pose significant challenges, with ChartQA requiring advanced reasoning skills, exemplified by GPT-4's (OpenAI, 2023) chain-of-thought approach specifically tailored for it. Furthermore, InfoVQA necessitates a thorough understanding of large images' content. DocVQA and InfoVQA are evaluated via the official competition leaderboard[2] with confidential test sets, complying with recent VDU literatures like Donut (Kim et al., 2022) and Pix2Struct (Lee et al., 2022). ChartQA has a public test set and is evaluated with an exact-match-based accuracy, as conducted by previous literatures (Lee et al., 2022; Masry et al., 2022). Throughout the evaluation process, we assess all models under the real-world scenario, meaning that **we do not utilize ground truth OCR during the testing phase**. Instead, we rely on off-the-shelf detectors, which may contain some errors.

**Off-the-Shelf Detectors** For OCR, we employ CLOVA OCR API[3], while for general object detection, we utilize OWL-ViT[4] from Minderer et al. (2022). The MS-COCO dataset (Lin et al., 2014) supplies the 80 class labels required for semantic class label texts. More details on the detectors and label texts can be found in Appendix A.2.

**Environment and Hyperparameters** We summarize task proportions during training in Table 2. The training starts with a batch size of 384, a fixed learning rate of 1e-4, for 220K steps. Next, we adjust the batch proportion and hyperparameters for another 275K steps; a batch size of 96, and a learning rate of 5e-5 with a decaying scheduling. The LLM integration employs a batch size of 192, 50K steps with 0.5 GPU days with 32 A100 GPUs, and a cosine-scheduled learning rate of 1e-4. Other major hyperparameters are $(\lambda, \tau, d, d^*, k, d', n, \hat{n}, l) = (0.5, 0.07, 1024, 128, 224, 4096, 3072, 300, 90)$.

### 4.2 Results

Table 3 presents the performance of various Frozen LLM integration models on text-rich VQA benchmarks. Our proposed Cream integration demonstrates significant improvements compared to other Frozen LLM integrations, particularly in benchmarks that require advanced visual understanding capabilities. A notable characteristic of Cream integration is its fixed-size soft visual prompt usage, which remains constant at 224 tokens regardless of the number of texts within the image. This efficiency-oriented approach contrasts with methods that input all OCR tokens into the LLM. Consequently, our model does not rely on exceedingly large token lengths (denoted as |OCR|) to process document information, thereby increasing efficiency. Figure 7 illustrates the non-negligible size of |OCR|, which leads to inefficiencies in conventional techniques. Table 3 demonstrates Cream's superior performance over existing LLM integration methods, even in the absence of off-the-shelf detectors. In this setting, the inference speed was 0.25 sec/sample. In contrast, OCR-Vicuna7B (excluding OCR time) had a speed of 0.5 sec/sample. Further details can be found in Appendix A.2.

Table 4 presents standalone performance. The first group consists of state-of-the-art VDU models fine-tuned on each benchmark, with ChartQA's UDOP score obtained using official implementation and training settings (more details in Appendix A.4.3). The second group shows multi-task

[2] https://rrc.cvc.uab.es
[3] https://clova.ai/ocr/en
[4] https://huggingface.co/google/owlvit-large-patch14

| Model | Prompt Length | Use Auxiliary | ChartQA | InfoVQA | DocVQA |
|---|---|---|---|---|---|
| LLaVA-Vicuna7B (Liu et al., 2023a) | 256 | | 0.5 | 2.4 | 5.5 |
| LLaVA-Vicuna13B (Liu et al., 2023a) | 256 | | 1.4 | 3.1 | 5.9 |
| BLIP2-OPT6.7B (Li et al., 2023) | 32 | | 4.6 | 11.0 | 3.7 |
| BLIP2-FlanT5-11B (Li et al., 2023) | 32 | | 4.4 | 11.4 | 8.6 |
| Cream-Vicuna7B w/o off-the-shelf detectors | 224 | | **50.0** | **22.1** | **41.1** |
| OCR-Vicuna7B (Chiang et al., 2023) | \|OCR\| | ✓ | 6.2 | 13.6 | 29.2 |
| OCR-Vicuna13B (Chiang et al., 2023) | \|OCR\| | ✓ | 3.7 | 23.7 | 31.4 |
| OCR-GPT3.5 | \|OCR\| | ✓ | 15.9 | 26.6 | 62.4 |
| OCR-GPT4 (OpenAI, 2023) | \|OCR\| | ✓ | 34.3 | 25.0 | 75.9 |
| BLIP2xOCR-OPT6.7B (Li et al., 2023) | 32+\|OCR\| | ✓ | 17.5 | 30.4 | 6.2 |
| BLIP2xOCR-FlanT5-11B (Li et al., 2023) | 32+\|OCR\| | ✓ | 18.6 | 36.6 | 63.8 |
| Cream-Vicuna7B (**Proposed**) | 224 | ✓ | **63.0** | **43.5** | **79.5** |

Table 3: **Experimental results for various models on visually-situated NLU tasks.** The term \|OCR\| indicates the number of tokens required to input OCR texts to the models. Cream, when integrated with the frozen Vicuna, significantly outperforms other LLM integrations with an efficient prompt length.

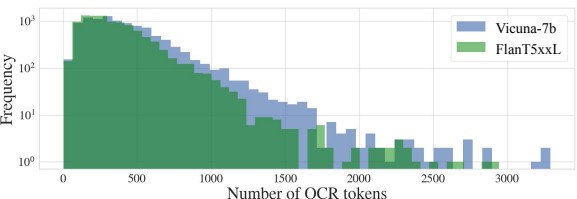

Figure 7: **Visualization of LLM token consumption induced by OCR.** We use DocVQA dataset (Tito et al., 2021) and tokenizers of Vicuna (Chiang et al., 2023) and FlanT5 (Chung et al., 2022).

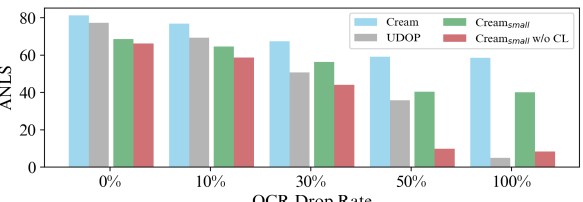

Figure 8: **Robustness in OCR deprivation.** We test model robustness by gradually increasing OCR drop rate on DocVQA samples. Cream shows robust performance even when no auxiliary information is given.

| Model | Size | Chart | Info | Doc |
|---|---|---|---|---|
| *Single-task Finetuned State-of-the-arts* | | | | |
| T5 (Raffel et al., 2020) | 0.8B | 59.8 | 36.7 | 70.4 |
| Donut (Kim et al., 2022) | 0.2B | 41.8 | 21.7 | 67.5 |
| MatCha (Liu et al., 2022) | 0.3B | **64.2** | 37.2 | 74.2 |
| Pix2Struct (Lee et al., 2022) | 1.3B | 58.6 | 40.0 | 76.6 |
| UDOP (Tang et al., 2022) | 0.7B | 60.7 | **47.4** | **84.7** |
| *Controlled Multi-task Model* | | | | |
| UDOP | 0.7B | 60.2 | **43.8** | 77.3 |
| Cream | 0.6B | **62.7** | 41.0 | **81.2** |

Table 4: **Standalone performance.** Cream shows comparable results to the recent VDU state-of-the-arts.

## 4.3 Analyses

We show some key findings in this section. Additional details and results are in Appendix A.1.

**Visual Prompt Length** Cream generates high-quality fixed-size visual prompts, reducing the computational cost associated with integrating OCR tokens into LLMs. The attention complexity per layer is $\mathcal{O}(\bar{n}^2 \bar{d})$, with $\bar{n}$ and $\bar{d}$ denoting the sequence length of tokens and the size of hidden dimension, respectively. Reducing the input token length ($\bar{n}$) significantly decreases the complexity in LLMs. Figure 7 showcases the substantial token consumption when incorporating OCR directly into LLMs. This underlines the potential computational advantages of Cream for visually-situated NLU tasks compared to other LLM integration strategies.

**Robustness on Auxiliary Information** Figure 8 illustrates Cream's high robustness against missing OCR in the text-rich DocVQA benchmark. We observed that Cream remained effective even without any OCR box input. Furthermore, CL significantly contributed to the increased robustness. We also

models, with Cream showing comparable performance to state-of-the-art models even when considering its multi-task setting and use of off-the-shelf detection results. Compared to the LLM integration, the standalone shows lower scores on challenging benchmarks, but higher scores in DocVQA. The LLM integration excels on questions requiring reasoning but struggles with those benefiting from direct image inspection. Including further analyses on the benefits of LLM integration, we discuss the proposed Cream's effectiveness and its robustness to OCR in the following analysis section.

| Model | DocVQA | | InfoVQA | |
|---|---|---|---|---|
| | Form | Table / List | Map | Arithmetic |
| Cream (Standalone) | **88.7** | **80.9** | **38.3** | 31.7 |
| Cream-Vicuna7B | 86.8 | 78.1 | 30.9 | **37.9** |

Table 5: **Comparative analysis between LLM Integration and Standalone.** The LLM Integration shows superior performance in tackling arithmetic problems.

| Model | Acc.↑ | ANLS↑ | nED↓ | BERT↑ | PPL↓ |
|---|---|---|---|---|---|
| BLIP2xOCR-FlanT5-11B | 18.6 | 24.7 | 67.1 | 76.3 | 24.5 |
| Cream-Vicuna7B | **63.0** | **60.3** | **31.4** | **91.0** | **2.0** |

Table 6: **Evaluation results with multiple metrics.** The models are assessed with the ChartQA benchmark using Accuracy, ANLS, nED, BERTScore, and PPL.

tested other publicly available OCR APIs/engines. Using a lightweight CPU-based OCR[5], Cream exhibited a smaller performance drop (-19%/-15%p) compared to UDOP (-29%/-20%p), demonstrating Cream's superior robustness. More detailed analyses are provided in Appendix A.1.1.

**Efficacy Analysis of LLM Integration** Table 5 shows comparative results between the LLM Integration and Standalone models. The assessment is concentrated on some categories which displayed conspicuous and statistically significant disparities in scores. The table clarifies that the Standalone Cream exhibits superior performance in scenarios that necessitate elementary key-value identification without the need for intricate reasoning, or tasks that involve comprehending extensive large map figures within an image. In contrast, the LLM Integration demonstrates higher competency in arithmetic problems that necessitate logical deduction.

**Evaluations with Diverse Metrics for VQA** For a comprehensive analysis of model performance, we utilize the test set of ChartQA benchmark and evaluate models using a series of diverse metrics. These encompass Average Normalized Levenshtein Similarity (ANLS) (Tito et al., 2021), Normalized Edit Distance (nED), BERTScore (Zhang et al., 2020), and Perplexity (PPL). This suite of metrics provides a well-rounded enhancement to the conventional exact-match accuracy, shedding light on various facets of model capabilities. The evaluation results are concisely summarized in Table 6. Notably, findings from ANLS and nED investigations depict a smaller performance gap than accuracy, yet unequivocally uphold the preeminence of the

---

[5] https://github.com/PaddlePaddle/PaddleOCR

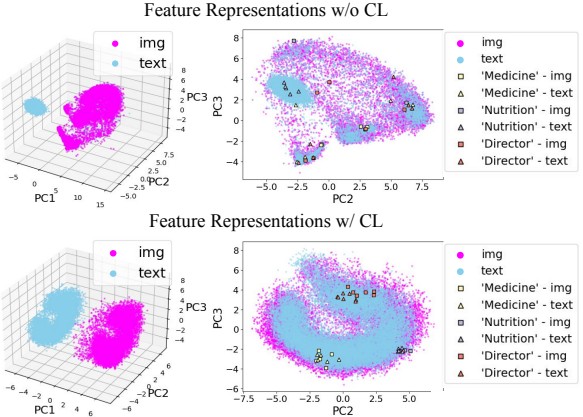

Figure 9: **Visualization of the common feature space through PCA.** The proposed CL aids in aligning features which possess similar semantics.

Cream model. Contrariwise, although most metrics underscore subpar performance for BLIP2, we find that its responses are not outrightly branded as nonsensical by the result of BERTScore.

**Feature Space Visualization** To further understand the role of CL, we conduct a visualization analysis on the common feature space. Figure 9 presents PCA results for the common feature space generated by the two encoders. It is evident that the CL-applied space more effectively removes the modality gap when excluding the first component. Employing the 2nd and 3rd components, we observed enhanced alignment and better clustering in the CL-applied space. These observations suggest that our CL leads to better-aligned embeddings from both encoders, significantly contributing to performance improvement. Further analyses can be found in Appendix A.1.2.

# 5 Conclusion

In this paper, we present **Cream**, a novel approach overcoming the constraints of current LVLMs for visual tasks on text-rich images. Cream's robust architecture synergizes a vision encoder, auxiliary encoder, and sophisticated techniques, including contrastive feature alignment. Our comprehensive evaluations confirm Cream's promising language-image understanding capabilities and robustness against OCR errors. The integration of Cream with LLMs provides a solid foundation for future improvements in visually-situated language comprehension. We believe our findings can easily be extended to other domains/tasks regarding visually-situated natural language understanding.

## Acknowledgements

The authors especially thank Seung Ho Choi, Jinbae Im, and members of NAVER Cloud Hyperscale AI Vision Understanding Team for helpful discussions and encouragement.

## Limitations

In this study, we have primarily focused on single-page image processing and successfully established a pioneering framework for integrating Cream with LLMs to address text-rich visual document understanding tasks. However, certain challenges and complexities associated with multi-page image analysis remain unexplored. Given the increasing demand for handling multiple images simultaneously, particularly in applications such as chatbot-like UIs where LLMs are commonly employed, extending our approach to multi-page processing represents a crucial aspect and calls for future research. Overcoming this limitation could involve distinct considerations, such as developing visual instruction data specifically tailored for multi-page images.

## Ethics Consideration

In our work, we inherit the ethical concerns of existing large-scale language models, such as data biases and privacy considerations. To mitigate these issues, we advocate for strict protocols during pre-training data curation, especially in public applications. Our model's pre-training uses controlled public data sources. Privacy-sensitive document processing, e.g., identification cards, requires diligent data handling practices for LLM development. Excluding such samples from training datasets is essential to prevent potential privacy breaches and unintended consequences. While our current approach relies on the autoregressive decoder's direct output, eliminating complex post-processing, it may be worth considering the investigation of post-processing techniques that address biases and privacy issues. This could provide an added layer of protection and ensure that model outputs adhere to the ethical guidelines within the field.

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

# A  Appendix

## A.1  Additional Analysis Results

### A.1.1  Performance Dependency of OCR Engines

In the pursuit of evaluating the models in realistic contexts, we present a set of supplementary experiments conducted using outputs from diverse, readily available OCR engines to assess model dependency on OCR. As demonstrated in the main manuscript, our proposed approach, Cream, displayed superior robustness when confronted with subpar OCR results. In this subsection, we offer further analytical outcomes produced by incorporating other prolific, public-access OCR engines. We not only tested CLOVA OCR API[6], but also

---

[6] https://clova.ai/ocr/en

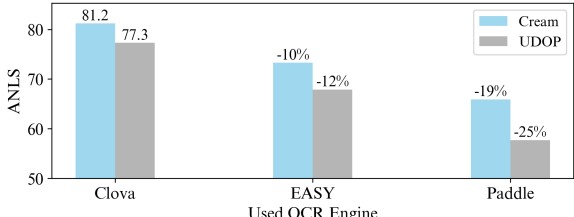

Figure 10: **Comparison of Cream and UDOP robustness across multiple OCR engines.** The performance of Cream and UDOP is assessed on DocVQA using CLOVA OCR, EasyOCR, and PaddleOCR engines. Cream displays better resilience, maintaining performance despite the varying capabilities of OCR engines.

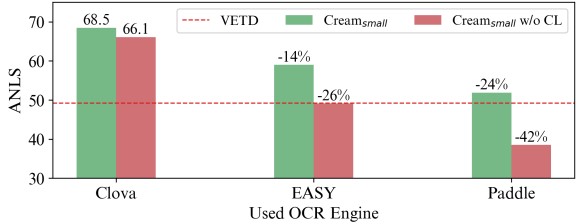

Figure 11: **Assessing CL's impact on robustness across multiple OCR engines.** We conduct evaluations on the DocVQA using CLOVA, EasyOCR, and PaddleOCR engines. The figure demonstrates that training with CL enhances robustness and mitigates performance degradation, even with varying OCR engine capabilities.

evaluated EasyOCR[7] and PaddleOCR[8]. Note that, PaddleOCR operates as a lightweight, CPU-based OCR engine, thus potentially acting as a preferred option for those seeking to decrease overall model deployment costs.

Figure 10 presents a comparative performance analysis of Cream and UDOP on the text-rich Document VQA benchmark, DocVQA. As can be observed in the figure, Cream exemplifies superior robustness. Figure 11, additionally, exhibits ablated models of Cream, evaluating the influence of CL. These findings are aligned with the trends noted in the main manuscript, underscoring that our implementation of CL improves robustness to OCR errors.

Figure 11 shows the performance of a new ablated model, Vision Encoder and Text Decoder (VETD), which is derived from $Cream_{Small}$ by removing the auxiliary encoder in the architecture. We use VETD as a performance baseline; if Cream's performance does not exceed that of

---
[7] https://github.com/JaidedAI/EasyOCR
[8] https://github.com/PaddlePaddle/PaddleOCR

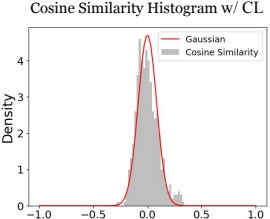 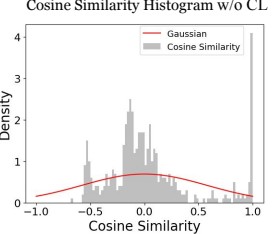

Figure 12: **Cosine similarity histograms.** The histogram of CL exhibits a distribution that closely resembles a Gaussian distribution.

| Model | ChartQA | DocVQA | InfoVQA |
|---|---|---|---|
| $Cream_{Small}$ | 56.7 | 68.5 | 30.7 |
| - w/ PaddleOCR at test | 53.1  (-6.4%) | 51.9 (-24.3%) | 25.5 (-16.9%) |
| - disable aux. at test | 41.1 (-27.5%) | 40.0 (-41.7%) | 13.4 (-56.3%) |
| $Cream_{Small}$ w/o CL | 54.6 | 66.1 | 31.3 |
| - w/ PaddleOCR at test | 47.3 (-13.3%) | 38.5 (-41.7%) | 25.1 (-19.8%) |
| - disable aux. at test | 10.6 (-80.6%) | 8.3 (-87.4%) | 13.1 (-58.1%) |
| VETD | 47.8 | 49.2 | 15.5 |

Table 7: **Ablation study results for $Cream_{Small}$.** The table shows that CL mitigates performance degradation due to the absence, or noise of auxiliary information (comparing blue values vs. red values). Note that PaddleOCR, as a CPU-based OCR engine, is lightweight but has lower performance.

VETD, it suggests that the addition of the auxiliary encoder may not be justified. As indicated in the figure, by incorporating the proposed CL technique, the overall enhancement in performance is observed, enabling $Cream_{Small}$ to outperform the baseline VETD, even with the deployment of a CPU-based lightweight OCR engine - PaddleOCR. Table 7 presents a detailed ablation study for $Cream_{Small}$. The robustness trend of CL against the noise or absence of auxiliary information can be observed across the entire evaluation datasets.

### A.1.2 Additional Analysis on the Common Feature Space with CL

Figure 12 illustrates the histograms of cosine similarity by calculating the cosine similarity between two randomly chosen embeddings within the shared common space. Under the assumption that the embedding space contains random (unit) vectors, a Gaussian distribution is expected for the histograms (Spruill, 2007). The red line depicted in the figures corresponds to the Gaussian distribution that minimizes KL divergence for each histogram distribution, specifically $\mathcal{N}(0, 1/140)$ for the CL condition and $\mathcal{N}(0, 1/3)$ for the non-CL condition. These results suggest that, with the integration of CL, the embeddings are distributed more

randomly or broadly across the embedding dimension, a factor that may significantly contribute to the enhanced performance of the Cream model.

### A.1.3 Analysis of Working Examples

Figures 13, 14, and 15 depict several working instances. Overall, the LLM integration model demonstrates a strong edge in tasks requiring arithmetic reasoning or prior knowledge. For instance, the final sample in Figure 13 and the first two samples in Figure 14 necessitate certain basic arithmetic computations. As also showcased with the quantitative evaluation in the main manuscript, the LLM integration model shows its capability for performing numerical operations among elements plotted in infographics and charts.

Conversely, the standalone model manifests certain advantages when the task demands basic text reading ability from text-heavy documents. We hypothesize this is due to the standalone model directly referencing the input image, whereas the integration model must encode all the information into the soft visual prompts (vectors). For instance, the instances in Figure 15 reveal that the LLM integration model either omits some letters or generates some characters inaccurately, particularly as the image's text size shrinks or becomes denser.

### A.2 Details on Off-the-Shelf Detectors and Inference Speed

Our preliminary studies on off-the-shelf OCR engines, as shown in Figure 10, led us to adopt CLOVA OCR API[9] for the experiments. On average, processing a sample from DocVQA using the above API took around 4 seconds. It's worth noting that, DocVQA often includes text-rich documents, leading to a relatively larger API time cost compared to other benchmarks. Although alternative lightweight OCR solutions like PaddleOCR[10] offered faster processing times, they revealed a noticeable quality gap, resulting in many text boxes being missed.

For general object detection, we employed OWL-ViT[11] from Minderer et al. (2022), using the MS-COCO 80 class labels[12] as the semantic class label texts for object detection. On average, the API took 0.66 seconds per sample for processing with

---

[9] https://clova.ai/ocr/en
[10] https://github.com/PaddlePaddle/PaddleOCR
[11] https://huggingface.co/google/owlvit-large-patch14
[12] https://gist.github.com/AruniRC/7b3dadd004da04c80198557db5da4bda

| Model | P1 | P2 | P3 | P4 | P5 |
|---|---|---|---|---|---|
| OCR-Vicuna7B | 25.6 | 19.2 | 20.1 | 6.4 | **28.3** |
| OCR-Vicuna13B | 28.2 | 24.8 | **29.5** | 7.5 | 29.0 |
| OCR-GPT3.5 | 50.1 | 60.4 | 47.9 | 17.9 | **60.5** |
| OCR-GPT4 | 52.1 | 63.8 | 60.2 | 30.9 | **70.4** |

Table 8: **Experimental results for OCR-LLMs in DocVQA with different prompt types.** The evaluation metric is ANLS, assessed after 500 samples from the validation set.

OWL-ViT$_{Large}$ and 0.22 seconds per sample with the OWL-ViT$_{Base}$. Our experiments revealed little performance difference between OWL-ViT$_{Large}$ and OWL-ViT$_{Base}$, suggesting that more efficient detectors could be tested. When examining the Cream standalone with only the object detector unavailable (OCR remained available), a minor performance drop in DocVQA was observed (from 81.2 to 80.9).

As highlighted above, off-the-shelf detectors can significantly influence a system's inference speed and deployment cost. Bearing this in mind, we designed our model to offer users the flexibility to choose options based on their specific needs. Factoring in all potential configurations, from disabling all off-the-shelf detectors to using each feature (without parallelism), our model can achieve inference speeds ranging from 0.25 to 4.91 (0.25 + 0.66 + 4) seconds per sample. Given these degrees of freedom, we anticipate that practitioners will be enabled to construct highly efficient systems using Cream, potentially with further enhancements such as parallelization.

### A.3 Considerations on LLM Prompts

### A.3.1 OCR-LLM Prompt Variations

We empirically observed LLMs' sensitivity to natural language prompts during testing. In order to evaluate the influence of various prompts on overall performance, we conducted a comparative analysis of five different types of prompts, all of which are evaluated through the text-rich DocVQA benchmark (Tito et al., 2021). As shown in Table 8, the choice of prompt significantly influences the overall performance.

Table 9 shows that integrating more specific conditions into the prompts generally results in improved performance. More specifically, we observed that text extraction tasks yield better outcomes when conditions specifically stipulate that responses should be derived from OCR tokens. Fur-

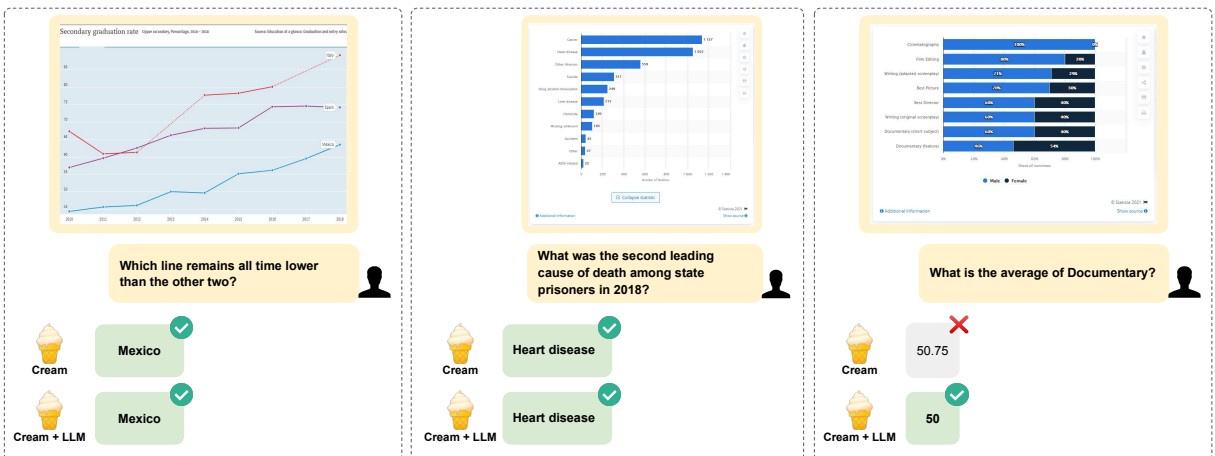

Figure 13: **Working examples on ChartQA benchmark.** Both models can handle various types of queries on chart images, but LLM demonstrates an advantage in queries that involve arithmetic reasoning.

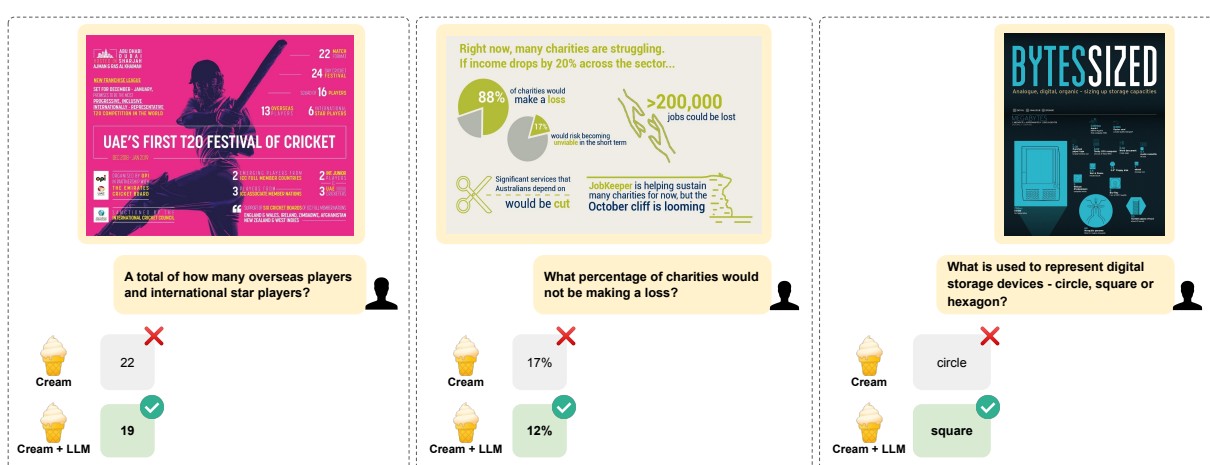

Figure 14: **Working examples on InfographicVQA benchmark.** Examples that require arithmetic reasoning or prior knowledge are shown. The LLM integration accurately calculates the sum of 13 and 6, resulting in "19" in the left example. In the middle example, it correctly performs subtraction to yield "12". In the right example, the LLM leverages prior knowledge about shapes to answer "square."

thermore, in line with the findings from Vicuna baselines (Chiang et al., 2023), implementing conditions such as "(please output answer text only)" effectively eliminates unnecessary sentences and words from the model's output, thereby enhancing the performance and precision. This was particularly crucial for achieving satisfactory performance on VQA benchmarks due to their reliance on edit-distance-based evaluation metrics.

In ChartQA, we found that imposing constraints on answer length and word count proved to be beneficial. We observed that constraining answer length and word count yielded favorable results. The addition of "Answer:" at the end of a prompt significantly assisted the model in executing the QA task. Furthermore, given that LLMs often generate questions as part of their responses, fur-

nishing a condition that excludes question-related text proved advantageous. Consistently, Prompt 5 (Table 9) exhibits the best overall performance and therefore was used to evaluate OCR-integrated LLM baselines. The results presented in Table 9 were obtained using 500 validation set samples from DocVQA.

It is worth noting that the results from OpenAI GPT APIs are specific to a certain version at a given time and should be considered during future replication efforts. Our experiments with GPT are conducted in May 2023. As OpenAI APIs actively evolve, updates might affect some trends and results.

We also examine concurrent work Latin-Prompt (Wang et al., 2023), which achieves notable performance in Document VQA benchmarks

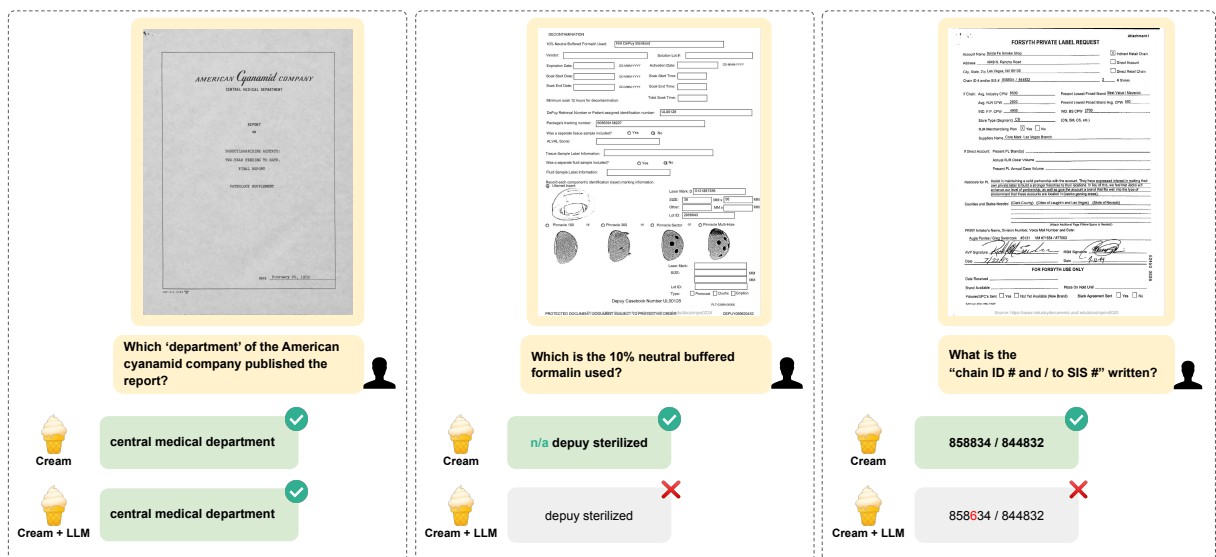

Figure 15: **Working examples on DocVQA benchmark.** The LLM integration model seems to overlook or generate some wrong characters when the text within the image becomes smaller and denser, compared to the standalone model.

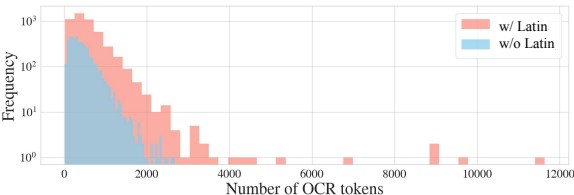

Figure 16: **Visualization of LLM token consumption induced by OCR according to the presence of Latin-Prompt.** The required OCR tokens (|OCR|) are displayed. We used the DocVQA and tokenizers of Vicuna (Chiang et al., 2023).

through a prompt engineering on LLMs. However, several specific conditions are required for Latin-Prompt to function effectively. Firstly, it requires text information of each line. In general, OCR recognizes text and bounding boxes in words, but some OCR APIs provide text and bounding boxes in lines. Latin-Prompt requires such line information. Secondly, using numerous spaces and indents to recover layout in OCR results increases input token length for LLMs, as depicted in Figure 16. Applying the method in Wang et al. (2023) entails higher computational costs due to the increased LLM tokens. When using Latin-Prompt with GPT-3.5, we record an ANLS score of 0.5724.

### A.3.2 Image-OCR-LLM Prompts

Table 10 showcases prompts for LVLMs tailored to perform a QA task given an image, OCR tokens, and a question. LLaVA (Liu et al., 2023a) begins

by processing a system message, which is followed by two conversation turns. Given that LLaVA is designed to generate detailed long output, including a brief answer example in the initial turn is beneficial.

For BLIP-2 (Li et al., 2023), we adhered to the original prompting rules, since they were already optimized for producing concise responses. When OCR is not utilized, we initially input the image, followed by the question to the model. Conversely, when using OCR, the image is input first, followed by the OCR texts, then lastly, the question. While BLIP-2 can answer questions relying solely on the image, the use of OCR was found to be essential for maintaining satisfactory performance.

### A.3.3 Cream Prompts

Table 11 displays the queries utilized for addressing individual tasks during Cream model training. To improve the model's generalization ability, we randomly sampled a variety of query types rather than using a single query for all tasks. The prompts in Cream training were designed as concise and straightforward sentences as a fundamental principle. Prompts for Captioning, QA, and QG tasks were adapted from BLIP-2 (Li et al., 2023).

### A.4 Additional Model Training Details

### A.4.1 Details of Contrastive Feature Alignment

As explained in Equation 1 in Section 3.3.3, we formulate a negative pair relationship even among

| No. | Prompt |
|---|---|
| 1 | Image OCR Result: {ocr tokens} / Question: {question} / 
 + (please output answer text only) 
 + (with no more than five words) 
 + Answer: |
| 2 | Image OCR Result: {ocr tokens} / Question: {question} / 
 + (please output answer text only) 
 + (Limit your answer to 50 characters or less) 
 + (Answers should not include question text) 
 + Exact Answer text in OCR Result: |
| 3 | Image OCR Result: {ocr tokens} / Question: {question} / 
 + (please output answer text only) 
 + (with no more than ten words) 
 + (Answer should not include question text) 
 + (The answer text must be included in the OCR text) 
 + Short Answer: |
| 4 | OCR tokens: {ocr tokens} {question} 
 OCR tokens: {ocr tokens} Question: {question} 
 OCR tokens: {ocr tokens} {question} A short answer to the question is 
 OCR tokens: {ocr tokens} Q: {question} A: 
 OCR tokens: {ocr tokens} Question: {question} Short answer: 
 OCR tokens: {ocr tokens} Given the image, answer the following question with no more than three words. {question} 
 OCR tokens: {ocr tokens} Based on the image, respond to this question with a short answer: {question}. Answer: 
 OCR tokens: {ocr tokens} Use the provided image to answer the question: {question} Provide your answer as short as possible: 
 OCR tokens: {ocr tokens} What is the answer to the following question? "{question}" 
 OCR tokens: {ocr tokens} The question "{question}" can be answered using the image. A short answer is |
| 5 | OCR tokens: {ocr tokens} / Question: {question} / 
 + (Please output answer text only) 
 + (With no more than 10 words) 
 + (The answer must be a word that exists within the OCR tokens.) 
 + Answer: |

Table 9: **Detailed examples of various inference prompts used in OCR-LLMs.** This prompts are used in the DocVQA (Tito et al., 2021) test, and **Prompt 4** are adapted from InstructBLIP (Dai et al., 2023).

in-modality features. This approach is rooted in two considerations: (i) even within the same modality, embeddings of different texts or objects in images should not carry identical meanings, and (ii) the quantity and the quality of negative pairs substantially impact the effectiveness of CL. When in-modality features act as negative samples, similarities should exist in the modality while maintaining differing semantics, thereby providing high-quality negative samples.

When selecting positive pairs, we select an image patch that encompasses the center point of each feature evidence box, and the initial token from the subword tokens of the corresponding auxiliary feature evidence (i.e., OCR texts or semantic labels from general objects). From these sampled positive pairs, we use all pairs that do not have the positive relationship to each other as negative pairs. We adopted the standard CL strategy where other samples in the mini-batch serve as negative examples, as depicted in Equation 1. This in-batch negative sampling tactic is widely-used in recent CL studies (Karpukhin et al., 2020; Radford et al., 2021b).

Undoubtedly, there is a potential to yield false negative samples, often a consequence of overlapping regions between negative and positive samples. Future research may focus on devising methods to filter out these inaccuracies from the objective. For instance, while sampling positive pairs, we may set a margin to prevent regional overlaps among chosen samples.

However, for simplicity, we opted for a straightforward tactic that computes the CL objective with a pair sampling strategy ($P_{uni}$ in Equation 1). Our experimental results suggest that the potential false negatives in Equation 1 are not a significant concern. Hence, as demonstrated in our paper, the suggested CL significantly enhances the model's learning process.

### A.4.2 Cream and LLM Integration Training Process

First, the standalone Cream model is independently trained, prior to its integration with LLMs. In the integration training, the weights of the standalone model are utilized as initial weights. The subsequent sections delve into the specific details of this training process.

| Model | Prompt |
|---|---|
| LLaVA (Liu et al., 2023a) | You are LLaVA, a large language model trained by UW Madison WAIV Lab. |
| | + You are able to understand the visual content that the user provides, |
| | + and answer userś question using image and natural language. |
| | + Follow the instructions carefully and provide answer |
| | + text only without question included, less than five words |
| | |
| | ###Human: What is the type of image? |
| | + (please output answer text only without question and explanation) |
| | + (with no more than five words) |
| | |
| | ###Assistant: The answer is a document image. |
| | |
| | ###Human: {question} {image} |
| | |
| | ###Assistant: |
| BLIP-2 (Li et al., 2023) | {image} Question: {question} Answer: |
| | {image} OCR tokens: {ocr tokens} Question: {question} Answer: |

Table 10: **Inference prompts for Image-OCR-LLM baselines.**

**Step 1: Training Standalone Cream** The model training commenced with a large batch size of 384, a fixed learning rate of 1e-4, and proceeded for 220K steps using 128 A100 GPUs. Although not compulsory, the large batch size expedited the loss convergence process. In order to gradually progress from simpler to more complex reasoning tasks at this phase, we emphasized text reading and masked text prediction tasks by assigning batch proportions of (TR, MTP, Capt., QA, QG) as (22%, 46%, 22%, 5%, 5%). Following this, we began the next phase.

During the subsequent phase, we modified the batch proportion and hyperparameters for an additional 275K steps: a batch size of 96, a learning rate of 5e-5 with a decaying schedule using 32 A100 GPUs, and increased the ratio of QA/QG tasks in the batch. Specifically, this phase was divided into two sub-phases to incrementally increase the QA proportions in the batch. Initially, the proportion (TR, MTP, Capt., QA, QG) was set to (7%, 14%, 26%, 48%, 5%), and the final 60K steps were executed exclusively with Document VQA datasets (QA 100%). The standalone Cream model training was completed in approximately three days.

**Step 2: Further Learning to Prompt LLMs** Once the standalone Cream model got trained, our focus shifted toward integrating it with LLMs by leveraging text-rich Document VQA datasets. We observed that the convergence of loss transpired more rapidly in comparison to the aforementioned standalone training, possibly due to both Cream and LLM being well trained. The LLM integra-

tion utilized a batch size of 192, proceeded for 50K steps, required 0.5 GPU days using 32 A100 GPUs, and employed a cosine-scheduled learning rate of 1e-4.

### A.4.3 Training Baseline Single-task and Multi-task UDOP

To train UDOP (Tang et al., 2022) under our settings, we used the official implementation and the guided training script obtained from the official GitHub repository[13]. Since the original paper did not test the ChartQA benchmark, we trained the model with it to obtain the corresponding result. During the UDOP training on ChartQA, we noted that the validation metric converged rapidly. After 20 epochs (equivalent to 25K steps), ChartQA's score began converging at 60.2. However, we trained it further, ultimately achieving a score of 60.7 at around 90K steps.

For the multi-task setting, we combined multiple datasets and trained the model for 115K steps until achieving a converged validation loss. In order to conduct our analysis under controlled conditions, we employed the same datasets utilized in the Cream's training. It is noteworthy that, although the original paper also reported results with an increased image resolution of 1024, the model weight, corresponding to the high-resolution training, was unfortunately not made publicly accessible. We instead used the available pre-trained UDOP with a resolution of 224. Although the high

---

[13]https://github.com/microsoft/i-Code/tree/main/i-Code-Doc

resolution could potentially contribute towards improved outcomes, as noted by Tang et al. (2022), the relatively marginal performance gap between the resolution settings implies the resolution of 224, which has demonstrated state-of-the-art performance, remains a compelling baseline.

### A.5 Details on Synthetic VQA Dataset

Drawing inspiration from recent VDU literature that utilizes unimodal QA benchmark datasets to augment model performance (Powalski et al., 2021; Tang et al., 2022), we extended unimodal datasets (Rajpurkar et al., 2018; Clark et al., 2020) by creating synthetic VQA datasets called Squad-VQA and TydiVQA. These were constructed by rendering context pages using WEBVICOB[14], a visual corpus generation tool based on HTML processing.

To boost the model's information extraction capabilities, we created another synthetic VQA dataset called Wikipedia Key-Value VQA (WKVVQA). WKVVQA consists of synthetic document images containing key-value pairs extracted from Wikipedia, as illustrated in Figures 5 and 17. WKVVQA documents contain key-value information and synthetic tables.

We generated WKVVQA through the following process. First, we extracted numerous key-value pairs from Wikipedia dump files by selecting HTML tables with either two rows or two columns. This basic strategy effectively gathers key-value data. For instance, `venue-EMNLP` and `year-2023` might be identified in a table with two columns. After gathering numerous key-value pairs, we filtered out rare keys and values based on their frequency. These procedures produced a large set of key-value pairs. The key-value pairs are then randomly plotted on white background images. Synthetic tables and card-like objects were automatically generated by simple rule-based manual algorithms.

### A.6 Complete Dataset Examples

Figure 17 displays example samples of the training datasets. We utilized an array of synthetic and real document images, as well as scene text and general images. While our primary focus lies on processing text-rich documents, incorporating diverse training data proved helpful since context-rich documents often contain figures or diagrams.

---

[14]https://github.com/clovaai/webvicob

## B Contribution of Authors

**Geewook Kim** led the project as a task force manager, initiated the project, and made decisions on overall progress while organizing the research paper. **Hodong Lee** managed the overall dataset construction, co-initiated the project, organized model evaluations, and significantly contributed to code development. **Daehee Kim** contributed to the model architecture with a focus on contrastive feature alignment and played a key role in crafting the manuscript. **Haeji Jung** managed dataset construction at the project's beginning, co-initiated the project, and contributed to its proof of concept. **Sanghee Park** handled data processing, evaluated off-the-shelf LLMs, and made substantial contributions to prompt engineering. **Yoonsik Kim** provided critical advice on research direction and development, heavily contributed to the manuscript, and participated in model architecture development. **Sangdoo Yun** shaped the overall research direction and contributed conceptualization of Cream as a senior researcher. **Taeho Kil** gave advice on overall research direction and development, and contributed to the manuscript as a senior researcher. **Bado Lee** advised the project from its beginning, co-initiated the project, and made significant contributions to creating the necessary environment and resources. **Seunghyun Park** advised the project from its inception, co-initiated the project, and significantly contributed to shaping the project's direction as a senior researcher.

All participants contributed to this manuscript.

| Task | Queries |
|------|---------|
| Text Reading | Read all texts. |
| | Read all texts in the image. |
| | Read all characters in the image. |
| | Given the image, read all texts. |
| | Given the image, read all characters. |
| MTP | Read masked texts. |
| | Read masked texts in the image. |
| | Given the image, read masked texts. |
| | Read all hidden texts that are covered by the mask area. |
| Captioning | Explain the image. |
| | Use a few words to illustrate what is happening in the picture. |
| | Using language, provide a short account of the image. |
| | Please provide a short depiction of the picture. |
| | Could you use a few words to describe what you perceive in the photo? |
| | Can you briefly explain what you see in the image? |
| | Briefly describe the content of the image. |
| | Provide a description of what is presented in the photo. |
| | Write a description for the photo. |
| | Write a short description for the image. |
| QA | {query} |
| | Q: {query} |
| | Question: {query} |
| | Given the image, answer the following question. {query} |
| | Based on the image, respond to this question with a short answer: {query}. |
| | Use the provided image to answer the question: {query}. Provide your answer as short as possible. |
| | What is the answer to the following question? "{query}" |
| | The question "{query}" can be answered using the image. |
| QG | Given the image, generate a question whose answer is: {answer}. |
| | Based on the image, provide a question with the answer: {answer}. |
| | Given the visual representation, create a question for which the answer is "{answer}". |
| | From the image provided, craft a question that leads to the reply: {answer}. |
| | Considering the picture, come up with a question where the answer is: {answer}. |
| | Taking the image into account, generate a question that has the answer: {answer}. |

Table 11: **Task-specific queries (prompts) used in Cream training.** The prompts for Captioning, QA, and QG tasks are adapted from BLIP-2 (Li et al., 2023).

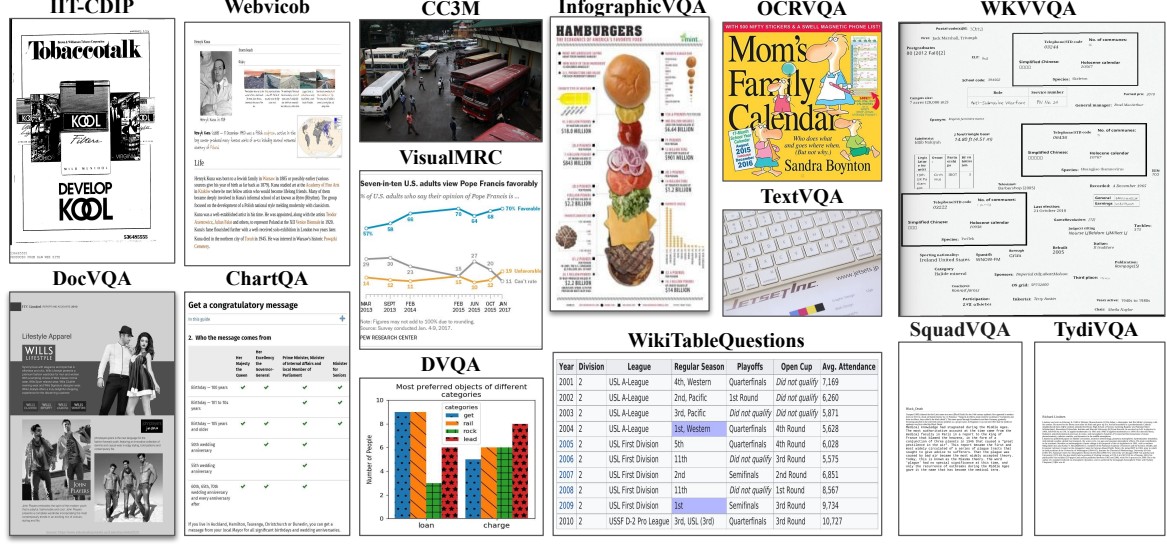

Figure 17: **Training Datasets.** This figure showcases samples from a variety of sources, including IIT-CDIP, WEBVICOB, Conceptual Captions (CC3M), and Document VQA benchmarks like DocVQA, ChartQA, VisualMRC, WikiTableQuestions (WTQ), OCRVQA, DVQA, and InfographicVQA. It also features samples from general VQA and scene text VQA datasets. Additionally, it presents samples from our new synthetic VQA datasets, namely Wikipedia Key-Value VQA (WKVVQA), SquadVQA, and TydiVQA.