# OpenReview forum: "Visually-Situated Natural Language Understanding with Contrastive Reading Model and Frozen Large Language Models"
_EMNLP/2023/Conference — EMNLP 2023 Main_

### Official Review · Reviewer_ZnEz · 2023-07-20

**Soundness:** 4

**Excitement:**

4: Strong: This paper deepens the understanding of some phenomenon or lowers the barriers to an existing research direction.

**Paper Topic And Main Contributions:**

This paper introduces a novel neural architecture called Contrastive Reading Model (Cream) that enhances the language-image understanding capability of Large Language Models (LLMs) in the visual domain. The main problem addressed by this paper is the limited application of LLMs in the visual domain, particularly in text-rich visual tasks. The authors propose Cream as a solution to bridge the gap between vision and language understanding and develop more sophisticated Document Intelligence Assistants.

The main contribution of this paper is the introduction of Cream, which combines vision and auxiliary encoders, fortified by a contrastive feature alignment technique, to achieve a more effective comprehension of language information in visually situated contexts within images. Cream overcomes the limitations of existing LVLMs by handling fine-grained features without missing image details. The authors demonstrate the compelling performance of Cream in visually-situated language understanding tasks and position it as a prominent model in the field of visual document understanding.

**Reasons To Accept:**

1. Novel Model: The paper introduces a novel neural architecture called Contrastive Reading Model (Cream). This new model has the potential to significantly improve the language-image understanding capability of LLMs in the visual domain, addressing a critical limitation of existing LVLMs.

2. Addressing a Practical Problem: The paper addresses the limited application of LLMs in the visual domain, particularly in text-rich visual tasks. By proposing Cream as a solution, the authors bridge the gap between vision and language understanding, leading to the development of more sophisticated Document Intelligence Assistants.

3. Experimental Results: The paper presents compelling experimental results demonstrating the strong performance of Cream in visually-situated language understanding tasks. This empirical evidence reinforces the effectiveness and practical applicability of the proposed model.

**Reasons To Reject:**

I think this is a good paper and should not be rejected.

**Reproducibility:**

4: Could mostly reproduce the results, but there may be some variation because of sample variance or minor variations in their interpretation of the protocol or method.

**Reviewer Confidence:**

4: Quite sure. I tried to check the important points carefully. It's unlikely, though conceivable, that I missed something that should affect my ratings.

---

> ### Author Rebuttal · Authors · 2023-08-29
>
> We appreciate the reviewer's time and effort in evaluating our paper and providing expert insights. The positive review of our paper is truly encouraging to the authors. We are grateful for the reviewer's valuable feedback and endorsement of its value. As we proceed with the final revision, we will diligently strive to address any areas of improvement and further enhance this work's value to the research community. Once again, we sincerely thank the reviewer for the reviewer's thoughtful evaluation.

---

### Official Review · Reviewer_h5KP · 2023-08-12

**Soundness:** 3

**Excitement:**

4: Strong: This paper deepens the understanding of some phenomenon or lowers the barriers to an existing research direction.

**Paper Topic And Main Contributions:**

The paper enhances Large Language Models (LLMs) for applications in the visual domain, particularly for understanding text-rich images. Recognizing the limitations of current models in this arena, the authors introduce the "Contrastive Reading Model (Cream)," a novel neural architecture. Cream integrates both vision and auxiliary encoders, reinforced by a contrastive feature alignment technique, to augment the comprehension of language information within visually situated contexts. This approach not only bridges the gap between vision and language understanding but also paves the way for sophisticated Document Intelligence Assistants. Through comprehensive evaluations, the paper underscores Cream's impressive performance, positioning it as a significant advancement in visual document understanding. The authors further contribute by sharing their codebase and datasets.

**Reasons To Accept:**

- The paper introduces the novel Contrastive Reading Model (Cream) with a unique methodological approach to enhance language-image understanding.
- The authors have made methodological strides in handling in-modality features and crafting positive and negative pairs, backed by rigorous evaluations across diverse visually-situated language understanding tasks.
- The analysis, coupled with a commitment to reproducibility by releasing their codebase, stands as a testament to the paper's thoroughness.

**Reasons To Reject:**

The paper falls short in providing an ablation study, which is crucial to elucidate the quantitative contributions of each introduced component. For example, even with the PCA analysis in Figure 9, the absence of a detailed examination of the contrastive learning technique leaves unanswered questions regarding its impact and significance.

**Reproducibility:**

4: Could mostly reproduce the results, but there may be some variation because of sample variance or minor variations in their interpretation of the protocol or method.

**Reviewer Confidence:**

3: Pretty sure, but there's a chance I missed something. Although I have a good feel for this area in general, I did not carefully check the paper's details, e.g., the math, experimental design, or novelty.

---

> ### Author Rebuttal · Authors · 2023-08-29
>
> We appreciate the reviewer for the efforts to provide valuable and insightful feedback. The reviewer's recognition of our methodological approach, rigorous evaluations across diverse visually-situated language understanding tasks, and our commitment to reproducibility greatly encourages us. We would like to address the raised concerns below.
>
> ### Ablation Study
>
> To address the weakness raised, we would like to draw attention to the ablation study presented in Figure 8, where the efficacy of Contrastive Learning (CL) is quantitatively examined under different levels of OCR noise. As demonstrated, a model utilizing CL shows significant robustness, especially in high OCR noise environments, highlighting the integral role that CL plays in improving resilience.
>
> Additional detailed analyses focusing on the effect of CL and OCR are provided in Appendix A.1.1 and A.1.2, Figure 11, and Table 5. These analyses further delineate the contributions of CL when Cream is subjected to OCR noise and auxiliary information noise.
>
> Regarding the PCA analysis in Figure 9, it offers a crucial supplementary view of our findings. While the practical difference brought by CL might not be explicitly visualized in the PCA, it provides an understanding of the resultant feature alignment in the representation space when CL is applied.
>
> In light of the reviewer's inputs, detailed discussions on CL and OCR in the Appendix will be relocated to the main text in the final revision for reader convenience. It is hoped that this response has clarified and enriched the reviewer's comprehension of the paper. As we undertake the final revision, we commit to addressing improvement areas and enhancing the value of this work for the broader research community. We extend our gratitude for the reviewer's valuable feedback in refining the work.

---

### Official Review · Reviewer_oQuR · 2023-08-13

**Soundness:** 3

**Excitement:**

4: Strong: This paper deepens the understanding of some phenomenon or lowers the barriers to an existing research direction.

**Paper Topic And Main Contributions:**

The paper proposes an information fusion approach to tackle the problem of visual question answering. Authors use multiple modalities (objects, labels, patches, OCR detections) to learn to answer questions about image content accurately. They propose a way to fuse different modalities, bridge them together and use this information to either directly predict an answer or generate a prompt used by an LLM to predict an answer. The paper describes an information fusion approach together with how it's used in model's training. Authors conclude the paper with evaluation against previous related approaches and some analyses showing that this information fusion approach works well.

**Questions For The Authors:**

1. Looking at examples in Figure 14 and 15 made me think about model's capabilities when questions are not asking about the most / the least / the specific type of information. Two examples in Figure 14 ask about the top country or a top item (US and PP). Does the model just really on spurious correlations in answering those? Can it answer 'which country is the third country in terms of the % of immigrants ...'? It should answer France. I think it is important to emphasise it in the paper, or a bit speculate if the model answers questions for the right reasons as the only metric you rely on is accuracy.

2. Lines 259 - 266: Yes, patches and encodings from auxiliary models do have overlapping information, but this information is presented in a different way. Because two models are different and work with different modalities. Is it correct? Then it means that there is no ''direct'' correspondence between features.

3. Also, how do you get the correspondence between image patches and boxes? I don’t get how do you know that visual patch and detected object map? Where does this correspondence come from? Is it annotated? To clarify, you are mapping visual features from visual transformer with features of OCR boxes and object text labels.

4. How did you choose negative examples? Do negative examples still have some sort of overlap with positive examples, e.g. negative patches might cover part of the object, albeit not object completely.

**Reasons To Accept:**

1. Interesting idea that is definitely relevant in multi-modal NLP: how to combine different information sources (multi-modality) and, perhaps, use it along the large LLM for some task. The datasets are challenging as they require some sort of reasoning over charts, etc., so this is something that needs more attention in the field.

2. Possibly plausible way to align features from different modalities.

3. The results hold across multiple datasets and (more or less novel) models and propose novel insights for those models.

**Reasons To Reject:**

1. Evaluation of the models is done with accuracy (I believe). Accuracy is known to be not that good for VQA tasks as it does not tell us whether models answer questions because of spurious correlation or for the right reasons. I think a different evaluation metric here is needed to inspect that (related to my first question to the authors in Questions For The Authors below). Perhaps, mean rank on different question types or perplexity would tell how bad the model is actually.

3. Also, as both proposed model and models that were proposed before are generally bad on this task (acc to Table 3), what are the exact differences between Cream and previous approaches? What makes Cream better? Is it its more modular nature? This was missing and I think this is crucial to frame the proposed idea.

2. Model description, training objectives and evaluation are hard to understand. Reading these descriptions is a bit complicated, I got the idea, but it took me several reads to fully grasp what was happening there (which implies that the whole approach is technically quite complex and involves many elements).

4. As I understand, when generating soft visual prompts, the LLM predicts an answer only on text produced by Cream. Although this text is generated based on multiple modalities, how big is the effect of LLM in answering the question correctly or incorrectly? Does it really really on the output of Cream or it just has a lot of knowledge about things and often outputs the correct answer (e.g., reporting bias). Plus, comparing Table 4 and 3, it seems like standalone performance or previous approaches is often better than Cream + Vicuna 7B. For example, 63.0 for ChartQA for Cream-Vicuna7B, but MatCha gets 64.2.

**Reproducibility:**

3: Could reproduce the results with some difficulty. The settings of parameters are underspecified or subjectively determined; the training/evaluation data are not widely available.

**Reviewer Confidence:**

4: Quite sure. I tried to check the important points carefully. It's unlikely, though conceivable, that I missed something that should affect my ratings.

---

> ### Author Rebuttal · Authors · 2023-08-29
>
> We appreciate the reviewer's detailed feedback and comments. In the following, we will address and clarify the weaknesses and questions raised by the reviewer.
>
> ---
>
> ### On Evaluation Metric for VQA
>
> We concur with the reviewer's emphasis on diversified metrics - one of the core challenges in the Visual Document Understanding (VDU) research community. We first briefly recap the VDU benchmarks and their metrics:
>
> 1. DocVQA [1] and InfographicVQA [2] have been proposed with diverse question types related to an image, which holistically appraise model capabilities. However, we cannot access the test split since they are evaluated via the official competition leaderboard [1, 2] with *confidential test sets*, complying with recent VDU literatures like Donut [3] and Pix2Struct [4].
> 2. ChartQA [5] has a public test set. We evaluate VDU models with an exact-match-based accuracy, as conducted by previous literatures [4, 5].
>
> We propose to include the following discussions on additional evaluation metrics for the ChartQA benchmark with an extended table:
>
> | Table R1    | CREAM-Vicuna | BLIP2xOCR-FlanT5xxL |
> |-------------|----------------|------------------------|
> | Accuracy    | **63.0**      | 18.6                   |
> | ANLS        | **60.3**      | 24.7                   |
> | nED         | **31.4**      | 67.1                   |
> | BERTScore   | **91.0**      | 76.3                   |
> | PPL         | **2.0**       | 24.5                   |
>
> We emphasize various metrics, including Average Normalized Levenshtein Similarity (ANLS) [1], Normalized Edit Distance (nED), BERTScore [6], and Perplexity (PPL), specified in the review, complimenting exact-match accuracy for model capability understanding. However, Mean Rank is unlikely to be applicable since there is no candidate set provided in the benchmark.
>
> ANLS and nED show a reduced performance gap, yet maintaining CREAM's superiority. Conversely, despite multiple metrics indicating BLIP2's poor performance, the BERTScore doesn't starkly classify its responses as nonsense.
>
>
> We aim to incorporate these discussions in the revised manuscript to thoroughly address the reviewer's points.
>
>
> ---
>
> ### On the Difference between Cream and Other LLM-Based Models
>
> From the perspective of input modality, models in Table 3 can be categorized into three groups:
>
> 1. Vision backbone with LLM: visual features (soft visual prompt) are used as inputs to the LLM
> 2. OCR with LLM: the OCR results from the image are utilized in text form by the LLM, as evidenced by OCR-GPT and OCR-Vicuna models
> 3. Both with LLM: both visual features and OCR results are employed
>
> The first two types of models were limited in that they use uni-modal information to address visual question answering on text-rich images. The third approach tackles the problem with using multi-modal information.
> Our Cream, featured in the third category, is unique due to its discrete auxiliary encoder for processing feature evidence, e.g., OCR texts (Figure 3).
> Its integration contrasts with models like BLIP-2, which directly inputs OCR texts into the LLM, resulting in substantial computational demands (Figure 7 and Section 4.2).
> Cream's advantage is further solidified by our proposed Contrastive Learning (CL) technique, which manages auxiliary information more effectively, displaying resilience against diverse OCR noises (Figure 8 and Section A.1.1).
>
> In response to the reviewer's comments, we will revise the manuscript to accentuate our model's distinctive attributes and superior performance.
>
> ---
>
> ### On the Concern over Model Description and Objectives
>
> The revised manuscript will extend clarity in Figures 2 and 3, detailing the model's architecture. Additionally, revisions will be made to simplify equations and notation for ease of understanding.
>
> ---
>
> ### On the Functionality of Soft Visual Prompts
>
> In order to rectify any potential misconceptions, we would like to elaborate on the function of the soft visual prompt. The soft prompts bypass textual conversion; they directly feed outputs from the vision module's output hidden vectors to LLM as a continuous input. Note that these prompts are not converted into text form.
>
> Regarding the doubt about not resorting to soft visual prompts (e.g., reporting bias), we affirm that while simple commonsensical queries might be answered independently by the LLM, complex queries necessitate the soft visual prompt (as exemplified in Figure 1). Note that, evidence of the efficacy of Cream's prompts is apparent in the performance discrepancy between OCR-Vicuna and Cream-Vicuna. It underscores that the soft visual prompt, as created through the proposed Cream model, imparts more valuable and effective information to the LLM compared to using OCR solely as a contextualizing prompt for the LLM.
>
> ---
>
> ### On the Model Performance
>
> Although Cream could not beat all the specialized models, it still holds benefits. MatCha excels in ChartQA but has poor performances in DocVQA and Infographic VQA, as seen in Table 4. On the other hand, Cream efficiently handles varied tasks and shows better OCR noise resistance than methods like UDOP in a controlled multi-task setting (See Figure 8). We underscore that score boosting is achieved without the need for post-processing or ensemble in our unified model. We believe our comprehensive model could lead to future LLM enhancements.
>
> Upon evaluating LLM integration's pros and cons, we noticed better logical reasoning performance but a slight decrease in handling simple OCR tasks or small text interpretation in large images. This is also discussed in Section A.1.3. Given the reviewer's feedback and insight, we would like to highlight these points in the manuscript's main body.
>
> ---
>
> Through this response, we hope to have addressed the listed weaknesses in the review. Subsequently, we will delve into the specific questions raised.
>
> ---
>
> ### Q1: Can the model answer non-specific questions without relying on spurious correlations?
>
> As you rightfully mentioned, it is paramount to evaluate models holistically, ensuring they generate accurate responses not merely on the basis of spurious correlations, but also draw on a deeper, inclusive comprehension. We can state that the proposed models indeed possess this capacity. The following examples are correctly predicted answers emphasizing our model's capability against spurious correlations:
> - Q: Which country is the {first / fourth} in terms of the percentage of Sub-Saharan African immigrants? A: {U.S. / France}
> - Q: Is the sum of the smallest two bars greater than the difference of the largest and 2nd largest bars? A: Yes
>
>
> In our research, we utilized representative benchmarks including ChartQA [5], InfographicVQA [2], and DocVQA [1]. These benchmarks encompass an array of queries, from relatively simple to complex in nature, like queries necessitating the identification of a second or third highest position or those necessitating arithmetic operations. Addressing these text-rich benchmarks cannot be achieved solely via reliance on high-frequency correlations.
>
>
> Upon considering the reviewer's comments, we are eager to enrich discussions on evaluation metrics with the augmented results as discussed earlier in the response to the point ``On Evaluation Metric for VQA’’ (Table R1 refers). Additionally, we would like to refine Figures 14 and 15 to better exemplify the models' capabilities.
>
> ---
>
> ### Q2: On correspondence between different modalities
> In response to the point, the CL objective doesn't indicate that two distinct features should have a direct relationship or identical vector values.
> The CL objective recognizes that certain feature pairs (i.e., positive pairs of patches and subwords) maintain more similarity to other features and incorporates this understanding into calculating the objective.
>
> In the paper, we found that the proposed CL objective significantly contributes to increasing the overall performance and robustness to OCR noise. That observation again emphasizes the merit of our approach. By bringing the Vision and OCR features into a subspace, we create an organic interaction between these modalities, promoting a holistic understanding of the given tasks.
>
> ---
>
> ### Q3: How is the correspondence determined in CL and how to prepare the correspondence?
>
> In CL, correspondence is established based on the position of a visual patch and its associated feature within an image. We make the connection of each visual patch and its feature from specific image regions, such as the bounding box of an OCR text.
>
> To prepare the detection results, we utilize off-the-shelf OCR and object detectors to produce pseudo labels if the original training dataset does not include them. This is a common practice in the VDU field [3]. Further information on our dataset samples is available in our supplementary materials. Details about the detectors used can be found in Section 4.1 and A.2.
>
> ---
>
> ### Q4: How were negative examples selected in CL and is there a chance for false negative samples?
>
> To generate negative examples, we utilized a conventional method in which the other samples in the mini-batch function as negative examples, as demonstrated in Equation 1 of our manuscript. This approach, also known as ``in-batch negative sampling’’ is commonly used in the field of CL as indicated in references [7, 8].
>
> Undoubtedly, there is the potential to yield false negative samples, often a consequence of overlapping regions between negative and positive samples. Future research may focus on devising methods to filter out these inaccuracies from the objective. For instance, while sampling positive pairs, we may set a margin to prevent regional overlaps among chosen samples.
>
> In this study, for simplicity, we opted for a straightforward tactic that computes the objective with a sampled subset of pairs instead of the entire set ($P_{\text{uni}}$ in Equation 1). Our empirical evidence suggests that most of the time, false negatives are not a significant concern in the objective computation when the full set isn't in use. As a result, as demonstrated in Figure 8, the proposed CL significantly boosts the model's learning process. Based on the feedback, we would like to elucidate CL's role further and incorporate these discussions into the revised manuscript.
>
> ---
>
> Our response and the suggested revisions will address the raised points and further enhance the quality of the paper. Additionally, we are open to further discussions and will gladly respond to any more queries or doubts raised. Considering the reviewer's inputs, we will eagerly revise our paper to contribute better to the research community.
>
> ---
>
> ### References
>
> [1] Tito et al. ``ICDAR 2021 Competition on Document Visual Question Answering.’’ ICDAR, 2021.
>
> [2] Mathew et al. ``Infographic VQA.’’ WACV, 2022.
>
> [3] Kim et al. ``OCR-Free Document Understanding Transformer.’’ ECCV, 2022.
>
> [4] Lee et al. ``Pix2Struct: Screenshot Parsing as Pretraining for Visual Language Understanding.’’ ICML, 2023.
>
> [5] Masry et al. ``ChartQA: A Benchmark for Question Answering about Charts with Visual and Logical Reasoning.’’ ACL Findings, 2022.
>
> [6] Zhang et al. ``BERTScore: Evaluating Text Generation with BERT.’’ ICLR, 2020.
>
> [7] Karpukhin et al. ``Dense Passage Retrieval for Open-Domain Question Answering.’’ EMNLP, 2020.
>
> [8] Radford et al. ``Learning Transferable Visual Models From Natural Language Supervision.’’ ICML, 2021.

---

### Meta-Review · Area_Chair_Lthe · 2023-09-18

**Recommendation:** 4

**Metareview:**

The authors introduce the Contrastive Reading Model (Cream) designed to enhance text-only LLMs for visually-situated language comprehension. In particular, Cream aligns visual features from both image patches and OCR texts through a contrastive objective. The aligned features are subsequently encoded as soft prompts, enabling LLMs to interpret visually-situated language found in documents, charts, and infographics.

The unanimous consensus among reviewers is that Cream offers a novel approach to integrating visually situated language into text-only LLMs. Although there were concerns regarding the rigor of the experimental setup and ablations, I believe the authors adequately addressed these in their rebuttal.

---

### Decision · Program_Chairs · 2023-10-07

**Decision:**

Accept-Main

**Comment:**

The authors introduce the Contrastive Reading Model (Cream) designed to enhance text-only LLMs for visually-situated language comprehension. In particular, Cream aligns visual features from both image patches and OCR texts through a contrastive objective. The aligned features are subsequently encoded as soft prompts, enabling LLMs to interpret visually-situated language found in documents, charts, and infographics.

The unanimous consensus among reviewers is that Cream offers a novel approach to integrating visually situated language into text-only LLMs. Although there were concerns regarding the rigor of the experimental setup and ablations, I believe the authors adequately addressed these in their rebuttal.